# Suitability and Modification of Different Renewable Materials as Feedstock for Sustainable Flame Retardants

**DOI:** 10.3390/molecules25215122

**Published:** 2020-11-04

**Authors:** Stefan Gebke, Katrin Thümmler, Rodolphe Sonnier, Sören Tech, Andre Wagenführ, Steffen Fischer

**Affiliations:** 1Institute of Plant and Wood Chemistry, Technische Universität Dresden, 01062 Dresden, Germany; stefan.gebke@forst.tu-dresden.de (S.G.); steffen.fischer@tu-dresden.de (S.F.); 2IMT—Mines Ales, Polymers Hybrids and Composites (PCH), 6 Avenue de Clavières, F-30319 Alès CEDEX, France; rodolphe.sonnier@mines-ales.fr; 3Wood and Fibre Material Technology, Technische Universität Dresden, 01062 Dresden, Germany; soeren.tech@tu-dresden.de (S.T.); andre.wagenfuehr@tu-dresden.de (A.W.)

**Keywords:** flame retardants, renewable materials, phosphate carbamate, wheat starch, wheat protein, xylan, tannin

## Abstract

Due to their chemical structure, conventional flame retardants are often toxic, barely biodegradable and consequently neither healthy nor environmentally friendly. Their use is therefore increasingly limited by regulations. For this reason, research on innovative flame retardants based on sustainable materials is the main focus of this work. Wheat starch, wheat protein, xylan and tannin were modified with phosphate salts in molten urea. The functionalization leads to the incorporation of phosphates (up to 48 wt.%) and nitrogen (up to 22 wt.%). The derivatives were applied on wood fibers and tested as flame retardants. The results indicate that these modified biopolymers can provide the same flame-retardant performances as commercial compounds currently used in the wood fiber industry. Besides, the flame retardancy smoldering effects may also be reduced compared to unmodified wood fibers depending on the used biopolymer. These results show that different biopolymers modified in phosphate/urea systems are a serious alternative to conventional flame retardants.

## 1. Introduction

Many currently used flame retardants have been criticized due to ecological and toxicological concerns. Although halogen additives are very effective, they are increasingly regulated. Under the influence of heat, high amounts of carcinogenic dioxins and corrosive gases can be released. Considering that most people die from smoke inhalation during a fire and not from exposure to fire, the proportion of halogenated flame retardants on the market is decreasing [1,2,3,4]. Boron compounds such as boric acid and zinc or sodium borate are well-known flame retardants for wood and cellulosic products [5,6]. But since 2010, boron compounds are classified as toxic for reproduction and listed as substance of very high concern in the REACH regulations. Ammonium often appears in combination with phosphates, sulfates and halogen salts. The protective effect is based on the endothermic release of ammonia that leads to cooling of the fire and dilution of the combustion gases. The desired effect, in case of fire, can even occur at high humidity and high indoor temperatures. This leads to the pollution of room air with ammonia. However, limit values for emissions are progressively reduced according to EU regulation [7]. This problem leads to a growing demand for green and environmentally friendly flame retardants based on renewable resources [8,9].

A promising approach for the production of flame retardants is the modification of biopolymers by common phosphating agents in molten urea [10,11]. This method allows the incorporation of the fire proofing elements phosphorus and nitrogen in one step synthesis onto the biopolymer using phosphoric acid, monoammonium phosphate (MAP, NH_4_H_2_PO_4_) or urea phosphate (CH_7_N_2_O_5_P). The reaction occurs solvent free. Thus far, mainly polysaccharides such as cellulose and starch have been functionalized [12]. The reaction products contain primarily phosphate esters. Degrees of substitution of 1.0 and above were achieved depending on conditions. This is high compared to heterogeneous reactions without urea which are in the range below 0.1. In addition, the formation of isocyanic acid from urea melting leads to the introduction of carbamate groups. During the reaction, crosslinking reactions may occur between the carbamate groups, resulting in poorly soluble compounds. Of course, these phosphate esters also contain ammonium as a counter ion. However, based on the molecular mass, they contain much less inorganic ammonium salts so that the ammonia emissions should be significantly reduced. Thus far, the products were previously discussed as water absorbing hydrogel for applications in pharmacy, food industry, cosmetics and agriculture. There is a lack of data on flame retardant additives processed and tested through this method [13]. It is largely unknown whether the additives are applicable for industrial applications, what biopolymer is best suited for the system and what protective effect they have.

First investigations using wheat starch provided promising results up to a larger scale. Flame retardants based on starch modified with phosphate/urea systems were tested as additives for wood fiber material. Their flame-retardant properties were extensively characterized with small burner and smoldering tests, thermogravimetric analysis, and cone calorimetry, as well as pyrolysis combustion flow calorimetry. Flame-retardant efficiency depends on the phosphate content, solubility, reaction time, and the additive form (physical mixture, uncleaned, or cleaned synthesis products). The application of well-soluble additives has a positive effect on fire and smoldering protection. The calorimetric measurements show a significant reduction in heat release rate compared to untreated wood fibers. It could be confirmed that the flame retardants act exclusively in the condensed phase. The investigations prove that starch-based flame retardants are suitable, representing an alternative to common flame retardants in the wood fiber industry [14]. Thus, further biopolymers were investigated in a similar manner to assess the influence of biopolymer nature. Wheat starch, wheat protein (weipro), xylan and tannin were chosen as model substances with different properties. Thus, the current study investigates modified biopolymers as fire retardant additives for wood fiber materials. For this purpose, FR based on wheat starch, wheat protein, xylan and tannin were applied to wood fibers. The fire behavior of the specimens was assessed with small burner test as well as pyrolysis-combustion flow calorimetry (PCFC) measurements and cone calorimetry. In addition, smoldering investigations were carried out. Besides, the smoldering behavior stood in the focus of these investigations, too. Smoldering (or glowing) is a slow oxidation of a solid fuel accompanied by temperature increase and smoke emission without open flame appearance. Smoldering is more difficult to suppress and detect than open fires which contributes to its dangerousness [15]. It occurs after flame vanishing at the end of burning and concerns many biobased materials, especially woods. In rare cases, flame retardants also have effective protection against smoldering. Sometimes flame retardants even promote this process [16,17].

The current study investigates various bio-based materials as a renewable source for FR. The aim is to identify if different biopolymers modified in the phosphate/urea system reach different flame-retardant performances. Additionally, these FR biopolymers are compared to industrial flame retardant (*Kappaflam T4/729*). In consideration of the technical implementation during the present work only uncleaned reaction products were applied.

## 2. Results

### 2.1. Raw Material Synthesis and Characterization of Flame Retardants

Basic material was an industrial wheat starch, because phosphat-carbamatation of starch is a well-known reaction [11,18] and was scaled up successfully [14]. Wheat protein has a high native nitrogen content of about 12%. Xylan as important and easily available hemicellulose served as further promising polymer. Tannin was selected as model substance for extractives. The modified biopolymers were compared with *Kappaflam T4/729* (KF) as conventional flame retardant. As phosphating agent, mono ammonium phosphate (MAP, NH_4_H_2_PO_4_) was used.

All syntheses were carried through in a kneader as described in part 4. Based on former results [14] for synthesis of FR using starch, wheat protein and xylan a molar ratio (AGU:MAP:Urea) of 1:3:4 proved to be optimum. For modification, tannin ratio was 1:1:4.

The effect of FR depends on phosphate and nitrogen content of the modified biopolymers. The values are given in Table 1.

Weipro, starch and xylan have high phosphate contents (45.6–48.2 wt%). For xylan, a high phosphate value was reached in a clearly shorter reaction time. Tannin exhibits around half this value. All these biopolymers also contain high content of nitrogen, which is often claimed as a FR element acting in synergy with phosphorus [19,20].

### 2.2. Thermal Analysis and Fire Behavior

#### 2.2.1. Thermal Analysis

The thermal degradation of wood fiber insulation material was described in a former work [14]. All flame retardants and the wood fiber insulation material were characterized using thermogravimetric analysis under air. The results are shown in Figure 1. 

The thermal degradation of untreated wood insulation material starts at 100 °C and is nearly completed at about 500 °C. Kappaflam remains stable until about 300 °C, but the decomposition is already finished below 500 °C, the residue is nearly 10%. The degradation of the modified biopolymers begins earlier (about 200 °C) than *Kappaflam* but it takes higher temperatures for a complete decomposition. The mass loss of xylan takes a long time (more than 50% residue until about 600 °C), weipro has the highest residual mass (>13%). Therefore, all the investigated modified polymers may be suitable as FR. 

#### 2.2.2. Fire Tests

Tests according to DIN EN ISO 11925-2 [21] have been used to define the reaction to fire performance. All wood samples with FR biopolymers have a biopolymer content of 10 wt%. For comparison purpose, untreated wood fibers, wood fibers with the native biopolymers and the commercial FR (KF–content 10 wt%) were used. KF is a sulfur-based FR and not directly comparable to the phosphate-based biopolymers. However, statements about the effectiveness can be made. 

The results of the fire test, as well as the theoretical phosphate and nitrogen values of the wood fibers are shown in Figure 2 and Table 2. The wood fibers without additives and with native wheat starch and xylan achieved the worst results and reached the maximum fire cone height of 20 cm. Native weipro and tannin decrease only slightly the cone height. It can be concluded that the native biopolymers had no (starch and xylan) or negligible (weipro and tannin) flame retardant effects. In contrast to native biopolymers, FR biopolymers allow to reach the 15 cm line or lower values.

Figure 3 shows that the performance depends on the phosphate content. Only KF-modified wood does not follow the same tendency. Modified weipro and xylan with high phosphate contents provide the lowest fire cone height (10.4 and 9.3 cm respectively). Despite a similar phosphate content, modified starch is slightly less efficient. Modified tannin contains less phosphate and exhibits a higher fire cone height (15 cm) than the other biopolymer based FR. KF provides a significant effect with a cone height as low as 10.7 cm despite a much lower phosphate content (0.7 wt%). This can probably be ascribed to the combination of sulfur and phosphorus; both are considered as key flame-retardant elements. 

#### 2.2.3. Pyrolysis Combustion Flow Calorimetry

Pyrolysis combustion flow calorimetry (PCFC) was used to characterize the flammability of unmodified wood fibers as well as wood fibers modified with commercial FR and FR biopolymers. The data are given in Table 3. The corresponding HRR curves are shown in Figure 4.

Unmodified wood fibers exhibit a PHRR of 128 W/g at 363 °C, a total heat release rate (THR) of 13.3 kJ/g, a residue content of 13.5% and a heat of complete combustion of 15.3 kJ/g. These values are typical for cellulose based materials and quite low compared to many synthetic polymers [22]. In contrast, all flame-retarded samples (including KF) exhibit a two-step decomposition. The first peak is the highest and occurs at low temperature (240–300 °C). This decrease in thermal stability is ascribed to the wood dehydration by phosphoric acid from the decomposition of phosphate moieties. The reduction in THR is due to the char promotion effect of the FR (from 13.5 to 34.0 wt%) but also to a decrease of heat of combustion. Indeed, the residual is enriched in carbon and the heat stored in the char is generally high. For many charring polymers, char composition is close to C_5_H_2_ [23]. The second PHRR is observed at higher temperature with low intensity (400–420 °C). This peak is usually rarely observed in cellulosic biomass. Sinc e wood is richer in lignin, it may be due to the decomposition of lignin. This step is more visible for FR samples because the decomposition of cellulose is shifted to lower temperatures [24,25].

It is useful to evaluate the effectiveness of the fire retardants according to the phosphate content [26,27]. As an example, THR is plotted versus phosphate content in Figure 5. As cone height (in DIN EN ISO 11925-2 test), THR decreases when phosphate content increases. KF is a sulfur-based flame retardant and contains only a small amount of phosphorus. Wood fibers modified with KF exhibit the lowest first pHRR. The second lowest pHRR is achieved by wheat protein, which also has the highest phosphate content. It also scores the highest residue, the lowest THR and HCC of all flame retardants including the KF reference. Tannin-based FR has higher THR and HCC and lower residue content. Starch and xylan based FR exhibit intermediate performances.

#### 2.2.4. Cone Calorimetry

Cone calorimeter tests were carried out on untreated wood fibers as well as wood fibers modified with commercial FR and modified biopolymers as FR. The results are summarized in Table 4 and shown in Figure 6. 

Unmodified wood fiber exhibits the highest Peak of Heat Release Rate (PHRR), almost 500 kW/m². The curve follows the typical shape of a non-charring sample [28]. After ignition at 31 s, the HRR curve increases fast to a slight shoulder. After the shoulder, HRR rises to its maximum at about 75 s. At this point, the surface starts to break up. Oxygen and heat penetrate into the center of the sample and the wood fiber starts to burn more efficiently [29]. After PHRR, the rate decreases quickly up to around 80 kW/m² and then much slower. This last step is due to thermo-oxidation of the residue when the flame is vanishing (pyrolysis turns from anaerobic to aerobic). The sample residue at the end is negligible. The wood fibers treated with FR biopolymers and commercial FR additive exhibit a much lower pHRR (209–233 kW/m²), TTI (15–21 s), THR (around 10 kJ/g) and a higher residue content (17–24%). The lower TTI values compared to unmodified reference can be assigned to the lower thermal stability as observed in PCFC. Compared to the unmodified wood fiber, the HRR profile looks different. The shoulder has changed to a small peak followed by a plateau. The plateau corresponds to the formation of a protective char layer, where oxygen and heat cannot penetrate so fast inside the sample [30]. The residue after HRR test is much more stable. This is common for materials with phosphorus-based FR. [31] Consequently, carbon rich fibers are still observed at the end and the residue remains black (see Figure 7, right side for wood fibers modified with FR weipro—residues are similar for other FR wood fibers). Nevertheless, the residue content is lower than in PCFC due to thermo-oxidation at the end of the test. Effective heat of combustion (EHC) was calculated for the flaming period (when HRR is higher than 80 kW/m²). For all samples, its value is close to the heat of combustion calculated in PCFC, evidencing that combustion is complete. In other words, FRs act only in condensed phase, as char promoter. Total Smoke Production (TSP) is a measure of the smoke opacity. Its value is 1.3 m² for unmodified wood fibers and significantly decreases in a similar way for all treated wood fibers (0.36–0.55 m²).

Overall, there are few differences between all modified wood fibers. FR tannin seems to perform worse (pHRR > 300 kW/m²) probably because the phosphate content is lower. Nevertheless, there is no difference in residue content. Overall, it can be concluded that FR biopolymers are as efficient as the commercial one.

### 2.3. Investigation of the Smoldering Behavior

Long-term smoldering usually correlates with high mass loss and low residue at similar ventilation conditions. Short smoldering times usually are an improvement, but can also be an indicator of intense and hot combustion. The maximum temperature of the smoldering process allows conclusions about the smoldering intensity. High temperatures can lead to a faster spread of the smoldering and also be an igniter for open fires. 

The influence of various additives on smoldering behavior was investigated by a method of Hagen et al. [32] Table 5 lists the experimental results. Because of the low density of the wood fiber filling (30 kg/m³), the distribution of the material inside the wire mesh box have a great influence on smoldering behavior. A large number of repetitions with large amounts of flame retardants or a higher density would be necessary to secure the results. The test should therefore only serve as an initial assessment for the smoldering performance.

The reference flame retardant KF significantly reduce the mass loss, time and maximum temperature of smoldering compared to untreated wood fiber. This is consistent, as sulfur and phosphorus are known to reduce smoldering [33,34,35]. The addition of native or FR biopolymers has various effect on mass loss and smoldering time. 

FR xylan has the greatest impact on the smoldering parameters. After short-time smoldering, a termination of the process resulting in a high residue and low maximum temperatures. Residue is even higher than for KF-modified wood. Improvements are less obvious for other native or FR biopolymers, depending on the considered parameters. 

## 3. Discussion

Phosphate and nitrogen groups are incorporated to improve the flame-retardant properties. The focus of this study was to compare the properties of FR based on different biopolymers. The contents of phosphates (measurement via ICP-OES) are directly related to the flame-retardant effects. Application of the modified biopolymers to wood fibers for testing the fire-retardant properties took place in accordance with the European standard DIN EN ISO 11925-2 [21], PCFC and cone calorimeter tests. It could be shown that the modified biopolymers yield similar results as commercial FR.

Different biopolymer-based flame retardants from phosphate/urea systems were tested for the first time as additives for wood fiber material. Their flame-retardant properties were extensively characterized using thermal analysis, small burner test, calorimetric measurements, pyrolysis combustion flow calorimetry (PCFC) and smoldering test. All flame retardants act as a char promoter, and their efficiency depends mainly on the phosphate content. Thus far, FR starch, FR xylan and FR weipro have been the most promising candidates for further investigation. They achieved similar fire protection while containing a high phosphate amount. With a significantly lower phosphate content, FR tannin exhibits a lower efficiency. The calorimetric measurements show a significant reduction in heat release rate compared to untreated wood fibers. 

Regarding smoldering behavior, the biopolymer nature may be influential. Indeed, xylan seems to perform better. Nevertheless, this observation requires further investigation and provides a perspective for a new class of smoldering inhibitors.

In summary, the investigations should contribute to the fact that plant based FRs not only remain theory in the literature, but find their way into practice. The results prove that biopolymer-based flame retardants from phosphate/urea systems are quite practicable and represent an alternative to common flame retardants in the wood fiber industry and for other applications. Except for smoldering, FR biopolymers are as efficient as commercial FR *Kappaflam*
*T4/729*. Further investigations should contain the optimization of the reaction conditions (phosphating agent, time, molar ratio) as well as the properties for application (solubility, storage suitability).

## 4. Materials and Methods

### 4.1. Materials

All raw materials used for the flame-retardant synthesis are commercially available and were used without further pretreatment. Wheat starch (Hamstarch A), wheat protein (Weipro) and xylan were provided by Jäckering Mühlen- und Nährmittelwerke GmbH (Hamm, Germany). Tannin (technical grade) was purchased from Fauth GmbH & Co KG (Mannheim, Germany). Urea and phosphate agents were obtained from Carl Roth GmbH + Co. KG (Karlsruhe, Germany). The commercial reference flame retardant *Kappaflam T4/729* (sulfur content 24 wt%) was produced by KAPP-CHEMIE GmbH & Co. KG (Bielefeld, Germany). Loose untreated softwood fibers from GUTEX Holzplattenfaserwerk H. Henselmann GmbH & Co. KG (Waldshut-Tiengen, Germany) were used as matrix material for the fire behavior characterization. 

### 4.2. Methods

#### 4.2.1. Synthesis

The synthesis was carried out on a laboratory universal mixing and kneading machine LUK 2.5 (Werner & Pfleiderer GmbH Maschinenfabrik, Stuttgart, Germany) with two crank-shaped moving Z-kneading blades. The starting materials for the synthesis were added without previous homogenization in powder form in the kneading trough. The modification took place in a molar ratio between AGU:MAP:urea 1:3:4 for starch, xylan and weipro and 1:1:4 for tannin. The kneading arm rotation speed was 45 rpm and the jacket temperature amounted 160 °C.

#### 4.2.2. Phosphate and Nitrogen Content

Phosphate content were analyzed by inductive coupled plasma spectroscopy ICP-OES CIROS CCD from SPECTRO Analytical Instruments at the Institute of Soil Sciences and Site Ecology (TU Dresden, Dresden, Germany). Prior to measurements, the dry samples were subjected to microwave assisted digestion based on DIN EN 13805:2002 [36]. 

Nitrogen content were conducted using the device vario EL III from Elementar Analysensysteme GmbH (Langenselbold, Germany). 

#### 4.2.3. Thermal Behavior 

Thermogravimetric investigations were carried out on a Netzsch STA 449 F5 Jupiter^®^ from NETZSCH-Gerätebau GmbH (Selb, Germany). For the measurements, about 30 mg of sample material were weighed into aluminium oxide ceramics crucibles. The samples were heated starting at 25 up to 900 °C at a heating rate of 10 K/min under synthetic air atmosphere (75 mL air/min). The evaluation and generation of the DTG curves was carried out with the Netzsch Proteus Thermal Analysis software.

#### 4.2.4. Application

The modified biopolymers were applied after the synthesis without cleaning on loose wood fibers. Therefore, they still contain unreacted educts and low molecular weight products. In each case 10 wt% of the modified biopolymers was added to the fibers. The FRs were applied by spraying with a commercially available pump spray bottle in the form of a 10% aqueous solution. Slightly soluble compounds were finely dispersed prior to spraying with a batch Ultraturrax T 25 basic from IKA^®^ Werke GmbH & Co. KG (Staufen, Germany). 

#### 4.2.5. Small Burner Test (DIN EN ISO 11925-2)

The flammability was tested on wood fibers in a small burner test based on DIN EN ISO 11925-2 as shown in Figure 8 [21]. The fiber material was filled in wire mesh boxes with a fiber density of 30 kg/m³. The samples were ignited for 15 s. The assessment of the flammability and flame propagation is based on the fire cone height, which was determined after 60 s test duration.

#### 4.2.6. Calorimetry Analyses

Calorimetric measurements were carried out at C2MA-IMT Mines Alès, in France. The cone calorimeter experiments were done according to standard ISO 5660 [37] with a heat flux of 35 kW/m². Pyrolysis-Combustion Flow Calorimetry (PCFC) was carried out under anaerobic pyrolysis with heating rate of 1 K/s up to 750 °C and combustion in excess of oxygen at 900 °C (to ensure complete oxidation of fuels). All samples were tested in duplicate.

#### 4.2.7. Smoldering Behavior

The smoldering behavior was investigated by a method of Hagen et al. [32]. Figure 9 shows the experimental equipment for the smoldering tests. The fibrous material was filled in upwardly opened wire mesh boxes (200 × 65 × 65 mm) with a target density of 30 kg/m³. The data were recorded with a logger OM-DAQPRO-5300 of the company OMEGA Engineering GmbH, Deckenpfronn, Germany with type K thermocouples (sample length 200 mm, sample width 1.5 mm) of the company JUMO GmbH & Co. KG, Fulda, Germany. To monitor the smoldering temperature curve, 6 thermocouples were locked at a distance of 4 cm in the center of the grid box. The ignition source used was a heating plate of the type Yellow MAQ HS from IKA^®^-Werke GmbH & CO. KG, Staufen, Germany.

## Figures and Tables

**Figure 1 molecules-25-05122-f001:**
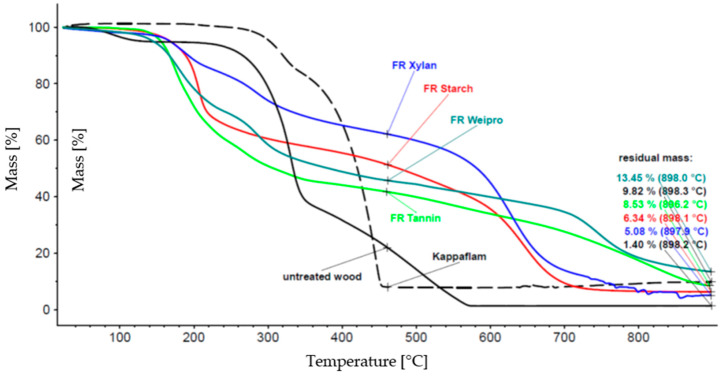
Thermal degradation of biopolymer-based FR compared to wood insulation material and *Kappaflam*.

**Figure 2 molecules-25-05122-f002:**
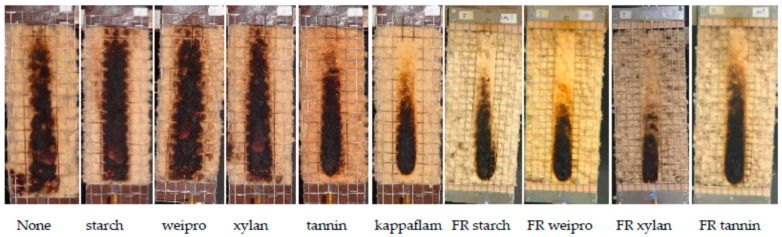
Results of fire test according DIN EN ISO 11925-2.

**Figure 3 molecules-25-05122-f003:**
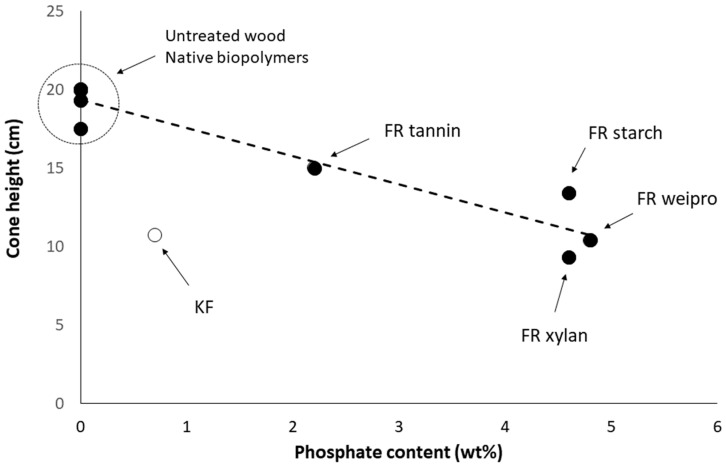
Cone height using fire test according DIN EN ISO 11925-2 versus phosphate content of FR-samples.

**Figure 4 molecules-25-05122-f004:**
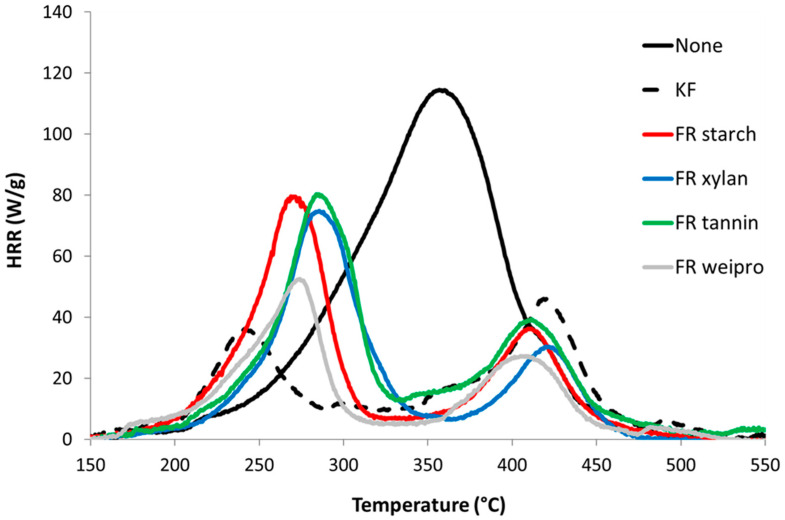
HRR curves for untreated wood fibers and fibers modified with FR in PCFC.

**Figure 5 molecules-25-05122-f005:**
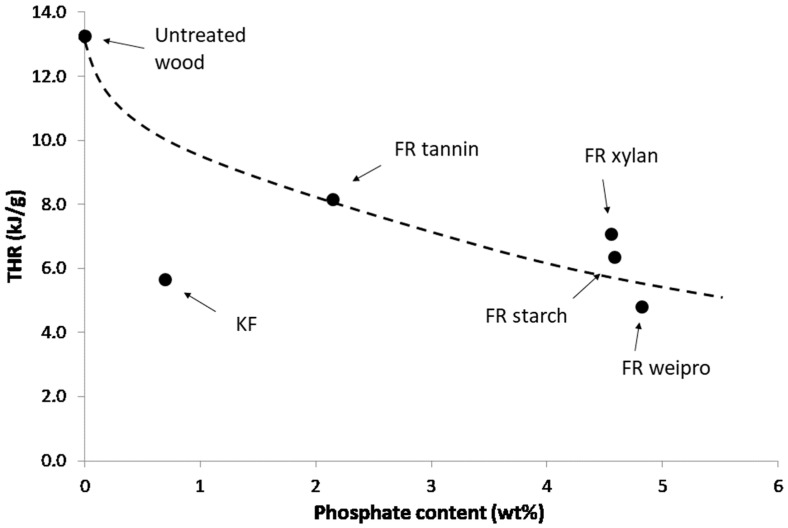
THR versus phosphate content for untreated wood fibers and fibers modified with FR in PCFC.

**Figure 6 molecules-25-05122-f006:**
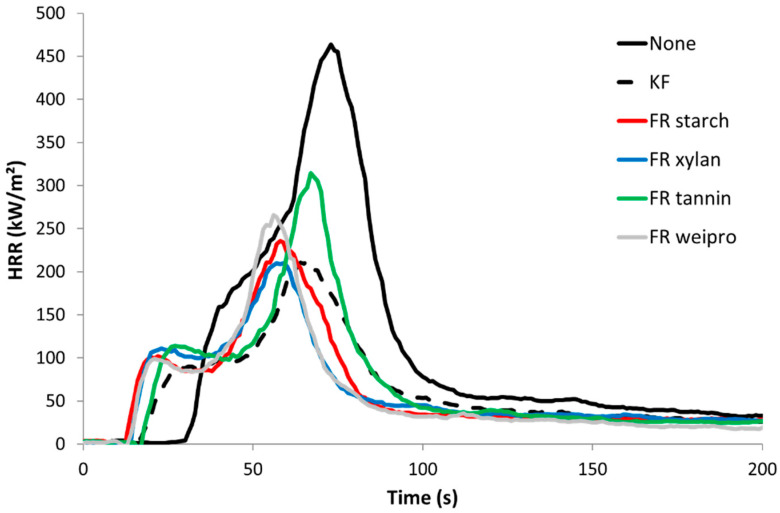
HRR curves for untreated wood fibers and fibers modified with FR in cone calorimeter (heat flux 35 kW/m^2^).

**Figure 7 molecules-25-05122-f007:**
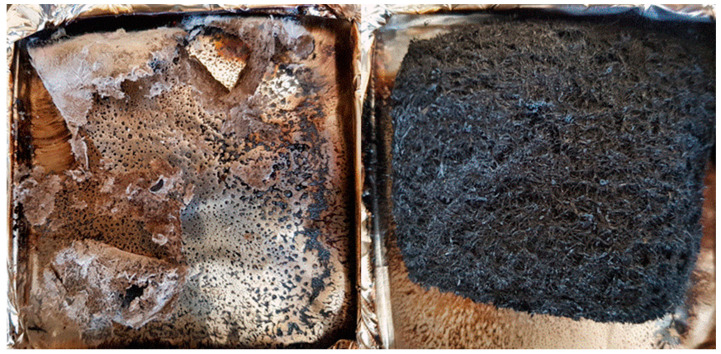
Residues from cone calorimeter for untreated wood fibers (left) and wood fibers with FR weipro (right).

**Figure 8 molecules-25-05122-f008:**
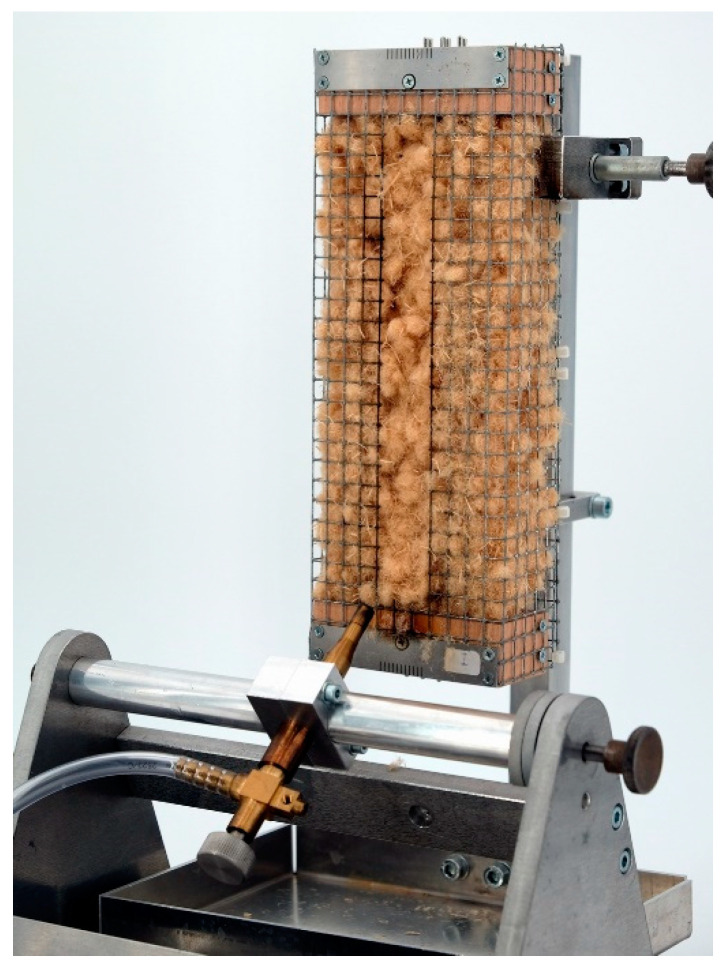
Equipment for single burner test according DIN EN ISO 11925-2.

**Figure 9 molecules-25-05122-f009:**
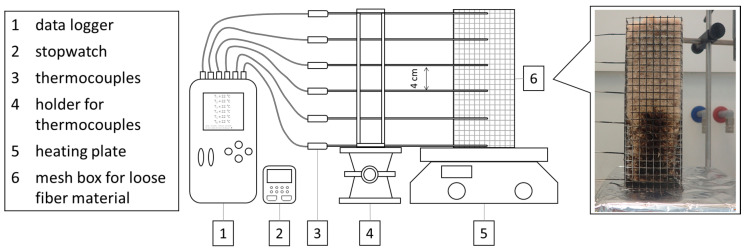
Experimental arrangement for the investigation of the smoldering behavior of loose wood fibers.

**Table 1 molecules-25-05122-t001:** Phosphate and nitrogen content of modified biopolymers.

Modified Biopolymer	Reaction Time [h]	PO_4_^3−^ [%]	N_tot_ [%]
starch	6	45.9	16.5
weipro	6	48.2	17.2
xylan	3	45.6	15.8
tannin	6	21.5	21.8

**Table 2 molecules-25-05122-t002:** Fire cone height, phosphate and (theoretical) nitrogen content of wood fibers with different FR.

Additive	Fire Cone Height [cm]	PO_4_^3−^ [%]	N_tot_ [%]
none	20.0	0	0
Native starch	20.0	0	0.1
Native weipro	19.3	0	1.3
Native xylan	20.0	0	0
Native tannin	17.5	0	0
Kappaflam	10.7	0.7	2.1
FR starch	13.4	4.6	1.6
FR weipro	10.4	4.8	1.7
FR xylan	9.3	4.6	1.6
FR tannin	15.0	2.2	2.2

**Table 3 molecules-25-05122-t003:** Results from PCFC in comparison with phosphate and nitrogen content.

Additive	T_1_ [°C]	P_1_HRR [W/g]	T_2_ [°C]	P_2_HRR [W/g]	THR [KJ/g]	Residual [%]	HCC [KJ/g]	PO_4_^3−^ [%]	N_tot_ [%]
none	363	128	-	-	13.3	13.5	15.3	0	0
KF *	241	35	418	40	5.7	30.2	8.1	0.7	2.1
starch	273	77	411	33	6.4	29.8	9.0	4.6	1.6
weipro	271	57	413	27	4.8	34.0	7.3	4.8	1.7
xylan	268	73	412	23	7.1	31.4	10.3	4.6	1.5
tannin	283	76	412	37	8.2	26.4	11.1	2.2	2.1

t_x_ time to Peak, P_X_HRR heat release rate at peak X, THR total heat release, HCC heat of complete combustion; * KF *Kappaflam T4/729* (sulfur-based commercial reference FR).

**Table 4 molecules-25-05122-t004:** Results from cone calorimeter measurements, test matrix wood fiber, additive content 10 wt%.

Additive	TTI [s]	PHRR [KW/m²]	THR [KJ/g]	Residual [%]	EHC [KJ/g]	TSP [m^2^]	PO_4_^3−^ [%]	N_tot_ [%]
none	31	486	15.9	2.5	16.3	1.3		
KF	19	209	10.3	22.9	13.3	0.36	0.7	2.1
starch	16	242	9.8	17.5	11.9	0.47	4.6	1.6
weipro	15	266	9.8	24.4	12.9	0.42	4.8	1.7
xylan	18	220	10.1	20.9	12.7	0.47	4.6	1.5
tannin	21	305	10.9	19.9	13.7	0.55	2.2	2.1

TTI time to ignition, PHRR peak heat release, THR total heat release, EHC effective heat release, TSP totalt smoke production.

**Table 5 molecules-25-05122-t005:** Results of investigations of the smoldering behavior.

Additive	Residue [%]	t_S_ [min]	T_S_^max^ [°C]	PO_4_^3−^ [%]
none	32.4	47.2	597	0
Native starch	36.5	36.5	600	0
Native weipro	46.0	36.2	611	0
Native xylan	26.7	27.1	506	0
Native tannin	22.3	51.2	572	0
Kappaflam	54.8	31.3	501	0.7
FR starch	38.4	36.2	669	4.6
FR weipro	36.3	34.9	701	4.8
FR xylan	67.4	25.7	576	4.6
FR tannin	51.6	31.0	557	2.2

t_s_ smoldering time, T_s_^max^ maximum smoldering temperature.

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
