# Peer review of "Suitability and Modification of Different Renewable Materials as Feedstock for Sustainable Flame Retardants"

_molecules, 2020, doi:10.3390/molecules25215122_

Round 1

Reviewer 1 Report

The paper from Gebke et al. reports on the modification of wheat starch, wheat protein, xylan and tannin with phosphates salts in molten urea, aiming at designing new sustainable flame retardants for wood. The idea proposed by the authors shows some novelty, but the conclusions should be better supported byy the experimental data. Some comments and suggestions are listed as follows:

  • it could be reasonable to perform TG (i.e thermogravimetric) analyses, either in nitrogen or in air, on the modified biopolymers
  • cone calorimetry: why did the authors use 35 kW/m2 heat flux instead of 50 kW/m2: indeed, the latter is the standard heat flux for wood materials; therefore, I would suggest to carry out the cone tests at 50 kW/m2. Besides, apart from the thermal parameters (collected in Table 4 - please note that kilo is k, not K! Please check also the other tables) a Table collecting the smoke parameters should be added to the revised text
  • it could be reasonable to perform some SEM-EDX analyses on the treated wood and on the residues after cone calorimetry tests, hence getting some information about the P and N content and distribution before and after the forced combustion test
  • the overall English needs some improvements

Author Response

  • We added the TG analysis.
  • The chapter “cone calorimetry” was revised.
  • We used SEM in [14], in this work we regarded only the theoretical P and N amounts.

Reviewer 2 Report

The manuscript presents several interesting results. It is well organized and the results are well explained and supported.

Innovative flame retardants based on sustainable materials such as:  wheat starch, wheat protein, xylan and tannin modified with phosphates salts in molten urea, is the focus of presented work.

The results on cone calorimeter and PCFC indicate that these modified biopolymers can provide the similar flame-retardant performances as commercial compounds currently used in the wood fibre industry.

These results show that different biopolymers modified in phosphate/urea systems are a serious alternative to conventional flame retardants.

I would suggest performing fire toxicity analysis to highlight the advantages of this new FR additives.

I consider this paper worth for publication in Molecules.

Author Response

Fire toxicity analysis were not possible in the frame of this work.

Reviewer 3 Report

The idea of the submitted paper is very interesting. Authors recognized the problem with conventional flame retardants and as a result of their idea offered the solution: innovative flame retardants based on sustainable materials. Commendable. Unfortunately, there are some issues that need to be addressed.

On the page 3, line 102 authors stated that …starch, wheat protein and xylan a molar ratio (AGU : MAP : Urea) of 1 : 3 : 4 proved to be optimum. Is this fact (conclusion) the result of their investigation or they used someone else's recommendation? On the same page in line 110-111 authors stated: “All these biopolymers also contain high content of nitrogen, which is often claimed as a FR element acting in synergy with phosphorus.” Author are advised to refer this statement to the reliable literature; like they done it on the page 10 in line 233-234. Likewise, page 5, line 152-153 sentence “This decrease in thermal stability is ascribed to the wood dehydration by phosphoric acid from the decomposition of phosphate moieties.” Some reliable literature or evidence (if it is authors conclusion) is needed to confirm this fact.

On the page 6, lines 156-160 authors discussed the second PHRR which they observed at higher temperature. Authors concluded that “While wood is richer in lignin, it may be due to the decomposition of lignin. This step is more visible for FR samples because the decomposition of cellulose is shifted to lower temperature [22, 23].” Why this conclusion hasn’t been confirmed by utilizing thermogravimetric analysis (alone or combined with IR or MS)?

Page 7, line 181-182, below the Table 4 the abbreviation PHRR - peak heat release should be written as heat release rate.

On the Fig. 5 authors showed residues from cone calorimeter for untreated wood fibers and wood fibers modified with FR weipro. Why authors didn’t show any photo from fire test which they have performed according to DIN EN ISO 11925-2?

Author Response

  • We selected the molar ratio based on our own investigations.
  • All further comments were made into account; the line numbers have changed now.
  • TG analysis and photos from fire tests were added.

Round 2

Reviewer 1 Report

The Authors have almost revised the manuscript according to the Reviewer's comments and suggestions. Now the manuscript seems suitable for publication.